# Maternal psychological distress and associated factors among pregnant women attending antenatal care at public hospitals, Ethiopia

**Getaneh Tesfaye[1], Derebe Madoro[2]\*, Light Tsegay[1]**

**1** Department of Psychiatry, College of Health Sciences, Axum University, Axum, Ethiopia, **2** Department of Psychiatry, College of Medicine and Health Sciences, Dilla University, Dilla, Ethiopia

\* derebemd@gmail.com

**Data Availability Statement:** Data cannot be shared because it may be replicated by others and because it may contain identifiable information. The

## Abstract

### Objective

Mothers who have endured psychological distress during pregnancy are more likely to have cognitive and behavioral issues for their baby, and are at greater risk for subsequent mental health problems for themselves. The aim of this study is to evaluate the prevalence of psychological distress during pregnancy in women attending antenatal clinics in Addis Ababa public hospitals and to find out if there are any associated factors.

### Methods

Hospital based cross sectional study was employed from May 7 to June 6, 2019 at public hospitals. A total of 810 pregnant women participated in the study selected through systematic random sampling technique. Kessler psychological distress Scale (K10) was used to measure psychological distress during pregnancy. Frequency tables and graphs were used to describe the study variable. The association between variables analyzed with bi-variable and multivariable binary logistic regression. A statistical significance was declared at p value < 0.05 with 95% confidence interval.

### Result

A total of 810 participants were included in the study with the response rate of 92%. The Prevalence of psychological distress among pregnant women was found to be 174(21.5%) with (95% CI, 18.6, 24.6). decreasing age [AOR = 3.61, 95%CI, 1.00, 13.01], no formal education [AOR = 3.57, 95%CI, 2.06, 6.19], having an abortion history [AOR = 2.23, 95%CI, 1.29, 3.87], having intimate partner violence [AOR = 4.06, 95%CI, 2.37, 6.94] and poor social support[AOR = 3.33, 95%CI, 1.95, 5.70] were statistically associated with psychological distress during pregnancy.

### Conclusion

This research found high prevalence of psychological distress during pregnancy compared with majorities of preceding studies. In this study we identified factors that are associated

data restriction was imposed by the Institutional Review Board. For further information, you can contact one of the committee members via this email address: seidshumye22@gmail.com. You can access the data upon request. On reasonable request, mobile: +251926023903, Dilla University, Dilla.

**Funding:** The author(s) received no specific funding for this work.

**Competing interests:** The authors have declared that no competing interests exist.

**Abbreviations:** AOR, Adjusted Odds Ratio; ASSIST, Alcohol, Smoking and Substance Involvement Screening Test; CI, Confidence Interval; COR, Crude Odds Ratio; HITS, Hurt Insult Threaten Scream; HPA, Hypothalamus Pituitary Axis; IPV, Intimate Partner Violence; SD, Standard Deviation; USA, United States of America.

with psychological distress in pregnancy. This includes, decreasing age, no formal education, having an abortion history, having intimate partner violence and poor social support. Psychological distress screening and potential risk factors for mental illness evaluations should be carried out during pregnancy for early diagnosis and intervention.

## Background

Psychological distress is a general term for stress, anxiety, and depression symptoms. High levels of psychological distress can be a sign of common mental illnesses like depression and anxiety disorders and are a sign of poor mental health [1]. Pregnancy is a time of physical, behavioral, and physiological transition. Pregnant women are also particularly vulnerable to psychological distress including anxiety, depression and stress [2, 3]. Psychosocial stressors are life experiences, such as the death of a family member, natural disasters, everyday hassles and job-related circumstances that allow the person who encounters them to cope with behaviors [4, 5]. Early years programming has been an important strategy for the prevention of developmental problems, largely influenced by greater understanding of environmental influences on the neuroplasticity of the young brain [6].

Accumulating research suggests that women are experiencing psychological distress during pregnancy. Psychological distress in the index of pregnancy is experienced in the range of 10–20 percent of pregnant women [7–10]. The social determinants influencing maternal health were poverty, low socio-economic status, decreasing age, low maternal educational status, abortion history, violence during pregnancy and single status are some of the factors for pregnancy with psychological distress [11–14]. The main mediating variables were location, maternal age at childbirth, parity, women's media exposure, and maternal health messages [15].

The occurrence of psychiatric disorders during pregnancy imposes a great burden on women and has the potential to negatively affect obstetric, fetal, and neonatal outcomes. Studies suggest that maternal psychological distress during pregnancy can have detrimental effects on the development of their child [16–20]. Maternal psychosocial stress can augment maternal inflammation and changes the hypothalamic-pituitary-adrenal (HPA)-axis related hormones. These changes consequently impact on the fetal neural development and involved in the etiopathogenesis of neurodevelopmental disorders of offspring. Particularly starting in mid-pregnancy, fetal growth can be affected by different aspects of maternal distress. Especially, children of prenatally anxious mothers seem to display impaired fetal growth patterns during pregnancy [21]. Evidence for this comes from many other studies including meta-analytic reviews established that psychological problems during pregnancy are associated with preterm birth and low birth weight [22–25]. Maternal psychological distress may also influence their offspring heart rate and triglycerides concentrations [26]. In addition, being psychologically distress also affects the maternal physical health. Large prospective cohort study observed associations of psychological distress with weight gain and elevated cortisol levels during pregnancy [27, 28]

The value of total lifetime costs of perinatal anxiety and depression combined is about £8500 per woman giving birth; for the United Kingdom, the aggregated costs were £6.6 billion [29]. Countries with low socio-economic position increases the adverse effect of negative life events on anxiety and depression during pregnancy [30]. Pregnant women in Low- and Middle-Income Countries including Ethiopia have been a neglected population in research on perinatal depression, anxiety and maternal and child outcomes [31].

Identifying pregnant women with significant psychosocial stress gives health care providers the opportunity to pay particular attention to these women and the associated risk factors. Reducing high stress and/or addressing associated risk factors can reduce the risk of adverse outcomes of the pregnancy [32].

Little is known about the epidemiology of psychiatric disorders especially psychological distress during pregnancy in Ethiopia with prior research focusing on the postpartum period and only in depression. In this study, we sought to evaluate the prevalence of psychological distress during pregnancy in women attending antenatal clinics in Addis Ababa public hospital and to find out if there are any associated factors.

## Materials and methods

### Study design and period

Hospital based cross sectional study was employed from May 7 to June 6, 2019.

### Study area

The study area was Addis Ababa, the capital and largest city of Ethiopia. It is the seat of the Ethiopian federal government. It was founded by Emperor Menellik II in the late 19th century. In Addis Ababa, there are 12 public hospitals providing health services of medical management, surgical intervention, obstetric and gynecological management, antenatal care, pediatric, orthopedic, psychiatric and other essential service for a large number of people. From those with 12 hospitals according to high antenatal care service follow up five hospitals were selected. This includes St. Paul hospital, Zewditu memorial hospital, Yekatit 12 hospital, Minelik hospital and Ras desta hospital.

### Source population

All pregnant women who had antenatal care visit in the public hospitals of Addis Ababa.

### Study population

Pregnant women who were visiting antenatal care clinic at five selected public hospitals during study period.

### Sample size determination

Sample size was determined by using Single population proportion formula.

$$n = Z_{\frac{\alpha}{2}}^2 \times \frac{p(1-p)}{d^2}$$

$$n = 1.96^2 \times \frac{0.25(1-0.25)}{0.03^2} \quad n = 800$$

Adding 10% non-response rate gives us a final sample size of 800

Where,

*n = Minimum sample size required for the study*

*Z = Standard normal distribution (Z = 1.96) with confidence interval of 95% and α = 0.05*

*P = Proportion of psychological distress 25% was taken from a study which is done in Addis Ababa, Ethiopia in antenatal depression among pregnant women* [33]

*d = Absolute precision or tolerable margin of error (d) = 3% = 0.03*

Concerning the sampling technique which was employed in the study was systematic random sampling. Study population come from selected five public hospitals; this include, St. Paul's Hospital Millennium Medical College, Zewditu memorial hospital, yekatit 12 hospital, Minilik hospital and Ras desta hospital. Study population was selected proportionally, from each hospital Fig 1.

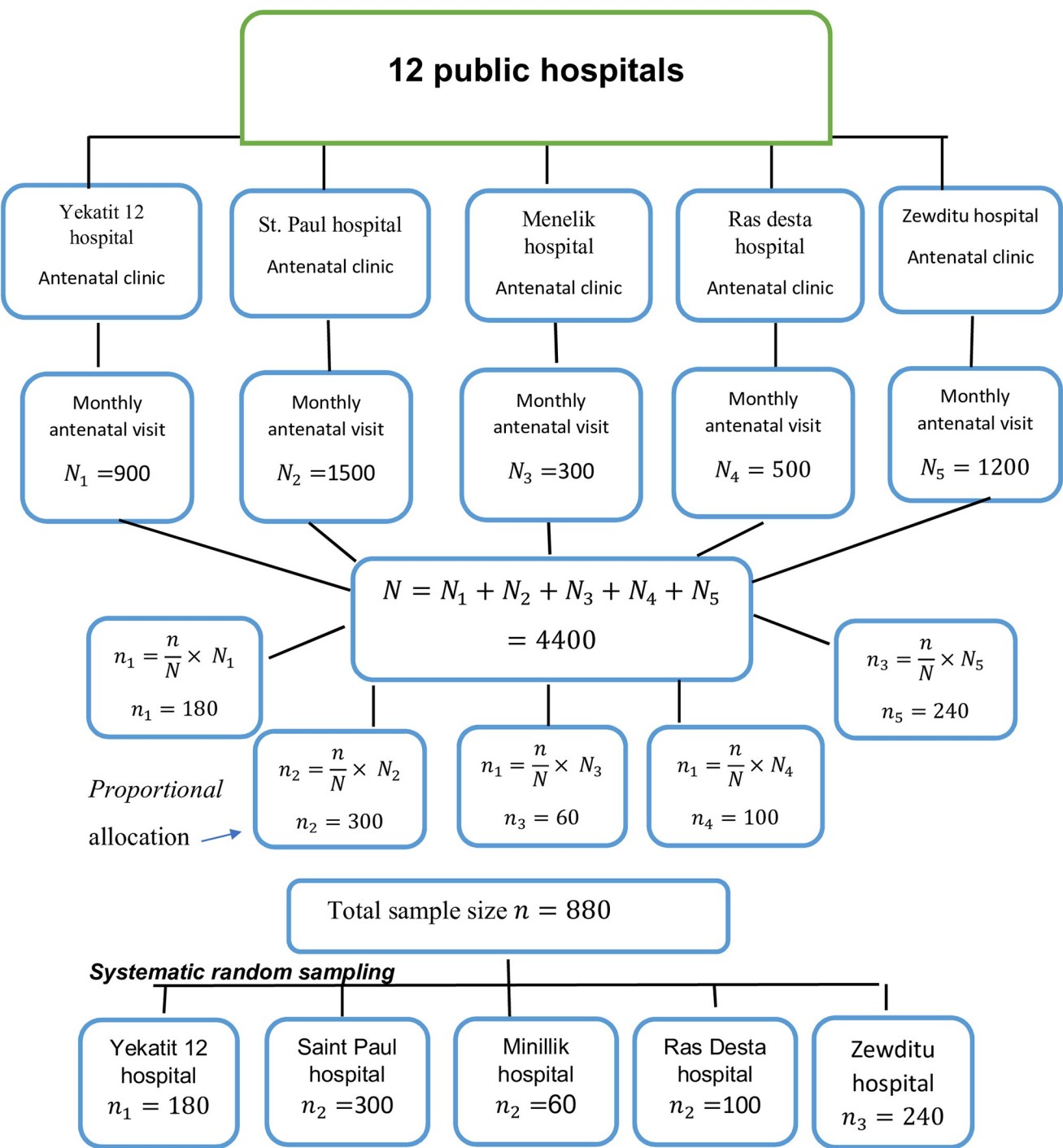

**Fig 1. Schematic presentation of sampling technique for the assessment of prevalence and associated factors of, psychological distress among pregnant women in public hospitals Addis Ababa, Ethiopia, 2019.**

Systematic random sampling was used to select study subjects from each hospital. The interval size (k) was calculated using the following formula.

$$k = \frac{N}{n}$$

$$k_1 = \frac{900}{180} = 5 \quad k_2 = \frac{1500}{300} = 5 \quad k_3 = \frac{300}{60} = 5 \quad k_4 = \frac{500}{100} = 5 \quad k_5 = \frac{1200}{240} = 5$$

$$k = 5$$

Therefore, the interval size for each hospital was 5. So that every five persons was selected from the study population.

Where *N*- Monthly population of selected hospitals

*n*- Sample size of each hospital (proportionally allocated)

## Measures

### Measures for the dependent variable (psychological distress)

Kessler Psychological Distress Scale: The Kessler psychological distress scale was developed by (L.S. Andersen, A. Grimsrud, D.R. William, L. Myeer, D.J. Eten, S. Seedat. It is a simple measure of psychological distress; the K10 scale involves 10 questions about emotional states, each with a five-level response scale. A score of 10–19 is likely to be well; 20–24 is likely to have a mild disorder; 25–29 is likely to have a moderate disorder, and 30–50 is likely to have severe distress K10 scale is a 10-item questionnaire that a person rating the 30 days anxiety and depressive symptoms experienced in a five-level likert [34]. Therefore, in this study, psychological distress was categorized as below 20 being "normal" and above 20 being positive for psychological distress. The K10 scale has already been validated in Ethiopia by Tesfaye et al. and yielded an excellent internal consistency of 0.93, sensitivity of 84.2%, and specificity of 77.8% at a cut-off point of 6/7 [35]. Thus, it was reasonable to use for this study population.

### Measures for the predictor variables

**Socio-demographic characteristics.** Semi-structured socio-demographic questionnaires were used to collect obstetric factors, and substance-related questionnaires were used to collect substance-related factors.

**Social support.** The Oslo-3 item social support scale is a 3-item questionnaire commonly used to assess social support and it has been used in several studies. The sum score scale ranges from 3–14, which has 3 categories: poor support (3–8), moderate support (9–11), and strong support (12–14) [36].

**Intimate partner violence.** Measured using the Hurt, Insult, Threaten, Scream (HITS) screening tool. During the HITS assessment, a provider asks a pregnant woman the following: How often does your partner physically hurt you, insult or talk down to you, threaten you with harm, and scream or curse at you? Each category is graded on a scale of 1 (never) to 5 (frequently), and a sum of all the categories is generated. A total score of 10 is suggestive of IPV [37].

**Substance use history.** According to Alcohol, Smoking and Substance Involvement Screening Test (ASSIST).

**Current use**: using at least one of a specific substance for non-medical purposes within the last three months (Alcohol, khat, tobacco, others).

**Ever use of substance**: using at least one of any specific substance for the non-medical purposes at least once in a lifetime (Alcohol, khat, tobacco, others) [38].

## Data collection procedures

Data was collected using face to face interview with interviewer administered questionnaire. The data was collected by 5 BSc. Female nurses, and supervised by two psychiatric nurses. Consequently, the entire data collection process had seven members. The nurses were employees of the hospitals. Accordingly, for each selected five hospitals there was one data collector and the supervisors were supervising them on each day. Training was provided to data collectors and supervisors for two days on methodology, ethical issue and how to administer questionnaires.

## Data quality control

The entire questionnaire was translated into local languages Amharic then it was translated back to English by an independent person to check for consistency and understandability of the tool. A back translation allows comparing translations with the original text for quality and accuracy. It also helps to evaluate equivalence of meaning between the source and target texts [39].

The questionnaire was pretested one week prior to the actual data collection on 5% of sample size at Addis Ketema Felege Meles health center in antenatal clinic and the questionnaire was checked for its clarity, simplicity, and understandability and items of questions was modified accordingly. Data collectors were supervised daily and the filled questionnaire was checked daily by the supervisors.

## Data processing and analysis

The collected data was checked visually for its completeness and the response was coded and entered into the computer using EPI data version 3.1, and then cleaned. The cleaned data was exported to SPSS Version 20 for analysis. Then the results were summarized and presented by tables, and charts. Furthermore, Percentage, frequency and mean were calculated. Firstly, bivariate binary logistic regression was performed to screen determinant factors of the outcome variable. Secondly, those predictor variables which were significantly associated with outcome variable with a p-value<0.25 in the bi-variable logistic regression analysis were entered into the multivariate logistic regression model for controlling the possible effect of confounders. The strength of the associated factors was presented by odds ratio with 95% confidence interval. The variables which have a statistical association were identified on the basis of p-values $\leq 0.05$. The model fitness for multivariate binary logistic regression was checked by using Hosmer and Lemeshow test.

## Ethics approval and consent to participate

Ethical clearance was obtained from ethical review committee office of Amanuel Mental Specialized Hospital, University of Gondar, College of medicine and health science with approval number of AM/146/5/107. Participants were first contacted with data collectors after permission letter has received from each hospital managers. Written informed consent was secured from each participant during study period. Participants were informed by the data collectors about the aim of the study and no names were recorded to maintain confidentiality. The participants were informed of their right to refuse or stop participating at any time during the interview. For some clinical outcome patients was linked to psychiatry support as necessary

and for the participants who were found problematic alcohol users, psychological distress positive during the study, communication to nearby psychiatric clinic was done in order to have further assessment on their condition. Confidentiality of respondents was maintained.

## Result

### Socio-demographic characteristics of the respondents

A total of 810 participants were included in the study with the response rate of 92%. The mean age (±SD) of the respondents was 27.21(±4.3), with age ranging from 18–43 years. Among the respondents, the highest age was in a range of 25–29 years 341(42.1%). Of the total participants about 564 (69.6%) were orthodox religion follower. The majority of the participants were married 720 (88.9%). The educational status of participants indicated that about 232(28.6%) of them are college and above and most of them were housewives 507(62.6%). Large numbers of respondents were from urban 758(93.6%) Table 1.

**Table 1. Distribution of participants by socio-demographic factors visiting antenatal clinics at public hospitals Addis Ababa, Ethiopia, 2019 (n = 810).**

| Variable | Frequency(N = 810) | Percent (%) |
|---|---|---|
| **Age** | | |
| 18–19 | 62 | 7.7 |
| 20–24 | 174 | 21.5 |
| 25–29 | 341 | 42.1 |
| 30–34 | 158 | 19.5 |
| 35–39 | 57 | 7.0 |
| 40 and above | 18 | 2.2 |
| **Religion** | | |
| Orthodox | 564 | 69.6 |
| Muslim | 146 | 18.0 |
| Protestant | 98 | 12.1 |
| Catholic | 2 | 0.2 |
| **Maternal educational status** | | |
| No formal education | 199 | 24.6 |
| Primary education | 131 | 16.2 |
| Secondary education | 171 | 21.1 |
| Preparatory | 77 | 9.5 |
| College and above | 232 | 28.6 |
| **Occupational status** | | |
| Farming | 20 | 2.5 |
| Merchant/private | 116 | 14.3 |
| Government employee | 167 | 20.6 |
| House wife | 507 | 62.6 |
| **Marital status** | | |
| Married | 720 | 88.9 |
| Not married | 62 | 7.7 |
| Divorced | 18 | 2.2 |
| Widowed | 10 | 1.2 |
| **Residence** | | |
| Urban | 758 | 93.6 |
| Rural | 52 | 6.4 |

**Table 2. Obstetric factor of the participant visiting antenatal clinics at public hospitals Addis Ababa, Ethiopia, 2019 (n = 810).**

| Variable | Frequency(N = 810) | Percent (%) |
|---|---|---|
| **Gestational age** | | |
| First trimester | 272 | 33.6 |
| Second trimester | 339 | 41.9 |
| Third trimester | 199 | 24.6 |
| **Parity** | | |
| Null para | 199 | 24.6 |
| Had one child | 244 | 30.1 |
| Had two and above children | 367 | 45.3 |
| **Pregnancy** | | |
| Planned | 678 | 83.7 |
| Unplanned | 132 | 16.3 |
| **History of abortion** | | |
| Yes | 101 | 12.5 |
| No | 709 | 87.5 |

## Obstetric characteristics of the respondents

During the study period, 239(31.5%), 330(43.5%) and 190(25%) subjects were in the first, second and third trimester of pregnancy, respectively. 352(46.4%) of the study subjects were multiparous and 642(84.6%) of the pregnancies were planned. Besides, history of abortion was experienced by 96(87.4%) of respondents Table 2.

## Maternal psychosocial characteristics

The current study revealed poor social support accounts 287(37.8%) and intimate partner violence reported by the respondents include 74(9.7%). Psychological distress was experienced by participants with poor social support and intimate partner violence was 108(13.3%) and 40 (4.9%) respectively.

## Substance use history

Majority of respondent had history of ever alcohol use 419(51.7%) but there is much decline in current alcohol use in pregnancy 297(36.7%). Among participants 32(4.0%) were current cigarette users in the past three months Fig 2.

## Prevalence of maternal psychological distress during pregnancy

Prevalence of psychological distress among pregnant women was found to be 174(21.5%) with (95% CI, 18.6–24.6).

## Factors associated with psychological distress among pregnant women

In bi-variable binary logistic analysis variables; decreasing age, no formal education, having an abortion history, current alcohol use, current cigarette use, having intimate partner violence, unplanned pregnancy, poor social support, and null parity were found to have p-value less than 0.25. Those variables fulfilled minimum requirement for further multivariate binary logistic regression.

From multivariate binary logistic regression only, variables; decreasing age, no formal education, having an abortion history, having intimate partner violence and poor social support,

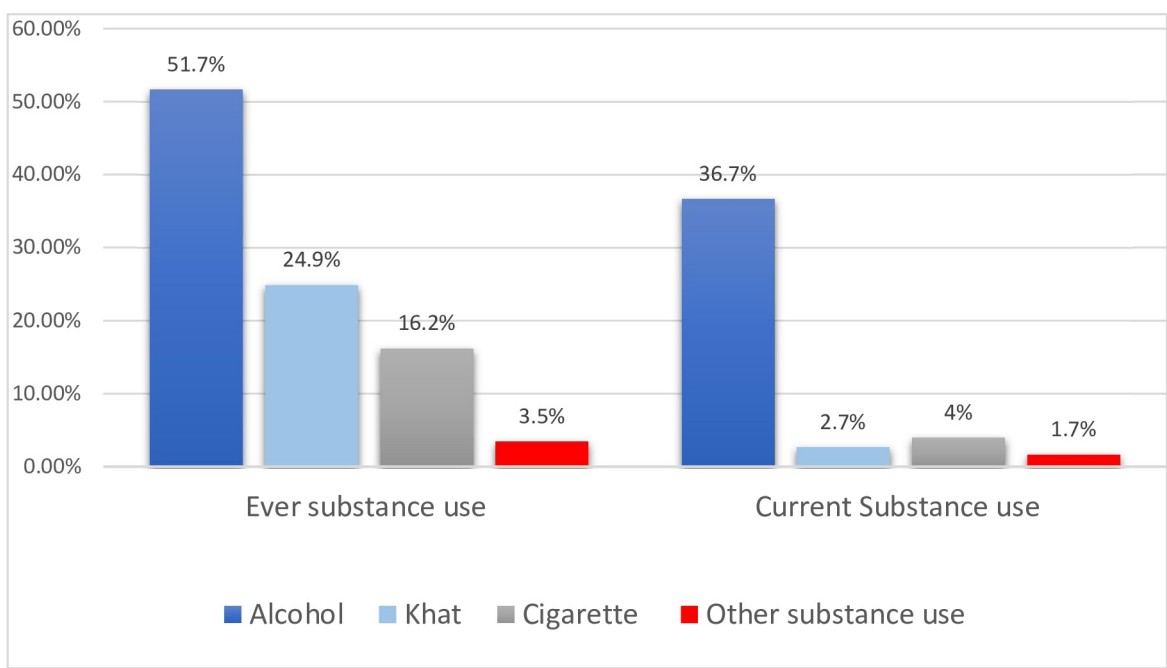

**Fig 2. Percentage of substance use history among participants visiting antenatal clinic at public hospitals Addis Ababa Ethiopia, 2019.** NB. Other substance use is (Cannabis, Amphetamine and opioids).

were statistically associated with psychological distress during pregnancy at p-value less than 0.05.

The odds of having psychological distress during pregnancy among respondents with age of 15–19 was 3.61 times higher as compared to those with age 40 and above [AOR = 3.61, 95% CI, 1.00, 13.01].

The odds of having psychological distress during pregnancy among respondents with no formal education was 3.57 times higher as compared to those with having educational level of college and above [AOR = 3.57, 95%CI, 2.06, 6.19].

Pregnant women who have an abortion history were 2.2 times more likely to have a psychological distress in pregnancy as compared to those with women who haven't an abortion history [AOR = 2.23, 95%CI, 1.29, 3.87].

The odds of having psychological distress during pregnancy among respondents who have an intimate partner violence was 4 times higher as compared to those with women who haven't an intimate partner violence [AOR = 4.06, 95%CI, 2.37, 6.94].

Pregnant women with poor social support were 3.3 times more likely to have a psychological distress during pregnancy as compared to those with strong social support [AOR = 3.33, 95%CI, 1.95, 5.70] Table 3.

## Discussion

This study identified the prevalence of psychological distress and its predictors of this behavior in Addis Ababa public hospitals.

The prevalence of psychological distress among pregnant women was found to be 174 (21.5%) with (95% CI, 18.6–24.6). The finding of the current study was similar with studies carried out in USA and Canada 21.2% [12], Bangladesh in pre-partum depressed women 18% [40] and another study in Ethiopia, depression among pregnant women found to be

**Table 3. Bi-variable and multivariable binary logistic regression analysis showing association between factors and psychological distress among pregnant women visiting antenatal clinics at public hospitals Addis Ababa Ethiopia, 2019(N = 810).**

| Explanatory variables | Psychological Distress | | COR, (95% CI) | AOR, (95%CI) |
|---|---|---|---|---|
| | Yes | No | | |
| Age | | | | |
| 15–19 | 35 | 27 | 3.37(1.07, 10.61) | **3.61(1.00, 13.01)** * |
| 20–24 | 27 | 147 | 0.47(0.15, 1.44) | 0.40(0.11, 1.38) |
| 25–29 | 63 | 278 | 0.58(0.20, 1.71) | 0.46(0.14, 1.53) |
| 30–34 | 28 | 130 | 0.56(0.18, 1.69) | 0.44(0.12, 1.52) |
| 35–39 | 16 | 41 | 1.01(0.31, 3.30) | 0.80(0.21, 3.02) |
| 40 and above | 4 | 13 | 1 | 1 |
| Educational status | | | | |
| Have no formal education | 71 | 128 | 2.92(2.03, 5.31) | **3.57(2.06, 6.19)** ** |
| Primary | 23 | 108 | 0.98(0.51, 1.86) | 1.19(0.62, 2.26) |
| Secondary | 29 | 142 | 1.25(0.71, 2.19) | 1.46(0.80, 2.68) |
| Preparatory | 14 | 63 | 1.36(0.68, 2.73) | 1.32(0.62, 2.83) |
| College and above | 37 | 195 | 1 | 1 |
| Pregnancy plan | | | | |
| Yes | 138 | 540 | 1 | 1 |
| No | 36 | 96 | 1.46(0.95, 2.24) | 1.27(0.78, 2.09) |
| Parity | | | | |
| Nulliparous | 53 | 146 | 1.43(0.95, 2.15) | 1.47(0.97, 2.35) |
| Primipara | 47 | 197 | 0.94(0.62, 1.42) | 0.75(0.46, 1.21) |
| Multipara | 74 | 293 | 1 | 1 |
| History of abortion | | | | |
| Yes | 30 | 71 | 1.65(1.04, 2.63) | **2.23(1.29, 3.87)** * |
| No | 144 | 565 | 1 | 1 |
| Current alcohol use | | | | |
| Yes | 71 | 226 | 1.25(0.88, 1.76) | 0.68(0.44, 1.06) |
| No | 103 | 410 | 1 | 1 |
| Intimate partner violence | | | | |
| Yes | 40 | 54 | 3.21(2.05, 5.04) | **4.06(2.37, 6.94)** ** |
| No | 134 | 582 | 1 | 1 |
| Social support | | | | |
| Poor | 108 | 215 | 3.96(2.49, 6.29) | **3.33(1.95, 5.70)** ** |
| Moderate | 39 | 208 | 0.14(0.87, 2.50) | 1.34(0.76, 2.38) |
| Strong | 27 | 213 | 1 | 1 |
| Current cigarette use | | | | |
| Yes | 10 | 22 | 1.70(0.79, 3.66) | 1.08(0.43, 2.71) |
| No | 164 | 614 | 1 | 1 |

NB. 1.00 reference

**p-value less than 0.001,

*p-value less than 0.05

The variables in AOR were identified on the basis of p-values ≤ 0.05.

Chi square = 8.71, df = 8, Hosmer lemshow test = 0.63

25% [33] However, the current study was less than the study was done in south Africa 26.5% [41], Pakistan 38.1% [42]. The possible reason for this difference might be variation in a single institution was used in South Africa and socio-economic variation. The other justification for this variation is that, In Pakistan the area is affected by conflict so that the psychological distress is high in that area. On the other hand, the finding of this study was higher than studies done in different studies in Netherland 7%, 8.1, 8.6% [27, 43, 44], USA 6.4% in first trimester women and 3.9% in third trimester women [12]. The possible difference with this study might be a population-based cohort study, prospective cohort study from early pregnancy in Netherlands.

Multivariate logistic regression revealed that decreasing age, no formal education, having an abortion history, having intimate partner violence and poor social support had association with psychological.

This study explored that decreasing age was associated with psychological distress. This observation is similar with studies done in USA [12], Pakistan [45]. Regarding age of onset, study revealed that with increasing age, psychological distress generally declined across the age range 20–64 years. The age of onset of most psychiatric disorders are early and also, psychological distress (anxiety and depression) in pregnancy [46].

No formal education is associated with psychological distress. This finding is similar with Brazil [47] and Bangladesh [40] studies. High educational levels seem to lead to lower levels of mental stress, later in life. To put it another way, education improves psychological health. That may be because more choices might made by educated people that they have greater control over their lives and better protection [48].

Having an abortion history also associated with psychological distress with pregnancy. This is supported by studies in Brazil [47], Pakistan [45]. Even though the link between abortion and mental illness is controversial, most women felt more regret, sadness and anger about the abortion than the pregnancy [49].

Intimate partner violence in this study associated with psychological distress in pregnancy. This finding is also revealed in Bangladesh [40], Pakistan [45]. One of the common psychological effects of physical IPV on women was identified as a mood disorder described as persistent or an episodic mood distortion. In several studies the physical IPV pregnant women were more likely to experience mood disorders [50].

Poor social support is also among the predictor variable for psychological distress during pregnancy. This finding is supported by studies done in Spain [51], increased perceived social support appeared to reduce the risk for antenatal anxiety in South Africa [52]. Social support provides physical and psychological advantages for people facing stressful physical and psycho-social events, and is considered a factor that reduces psychological distress when facing stressful events. In recent decades, numerous studies have been conducted on the impact of social support on health, quality of life and, in particular, mental health [53, 54].

## Conclusion and recommendations

This research found high prevalence of psychological distress during pregnancy compared with majorities of preceding studies. In this study we identified factors that are associated with psychological distress in pregnancy. This includes, decreasing age, no formal education, having an abortion history, having intimate partner violence and poor social support. Psychological distress screening and potential risk factors for mental illness evaluations should be carried out during pregnancy for early diagnosis and intervention.

## Limitations

We conducted a cross-sectional study, so relationships to causality cannot be determined. In addition, some of the data collection instruments were not previously validated in the target population.

## Supporting information

**S1 File.**
(RAR)

**S2 File.**
(RAR)

## Acknowledgments

I would like to express my thanks to University of Gondar College of medicine and health science, department of psychiatry and Amanuel mental specialized hospital for giving me the chance to conduct this study. I also would like to thank all pregnant women who were participated in this study.

## Author Contributions

**Conceptualization:** Getaneh Tesfaye.

**Data curation:** Derebe Madoro, Light Tsegay.

**Formal analysis:** Getaneh Tesfaye, Derebe Madoro, Light Tsegay.

**Investigation:** Getaneh Tesfaye, Derebe Madoro, Light Tsegay.

**Methodology:** Getaneh Tesfaye, Derebe Madoro, Light Tsegay.

**Supervision:** Derebe Madoro, Light Tsegay.

**Writing – original draft:** Getaneh Tesfaye, Derebe Madoro, Light Tsegay.

**Writing – review & editing:** Derebe Madoro, Light Tsegay.

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
