## [Decision Letter · Decision Letter 0]

21 Sep 2022

PONE-D-22-01497

Maternal psychological distress and its predictors among pregnant women attending antenatal care at public hospitals, Ethiopia

PLOS ONE

Dear Dr. Madoro,

Thank you for submitting your manuscript to PLOS ONE. After careful consideration, we feel that it has merit but does not fully meet PLOS ONE’s publication criteria as it currently stands. Therefore, we invite you to submit a revised version of the manuscript that addresses the points raised during the review process.

Both reviewers agree on the need to submit a revised version of your manuscript. From my point of view, it is relevant that you include a definition of psychological distress, as well as address the different aspects that the reviewers point out, especially in the discussion and limitations sections.

We look forward to receiving your revised manuscript.

Kind regards,

Patricia Moreno-Peral

Academic Editor

PLOS ONE

3.We note that you have stated that you will provide repository information for your data at acceptance. Should your manuscript be accepted for publication, we will hold it until you provide the relevant accession numbers or DOIs necessary to access your data. If you wish to make changes to your Data Availability statement, please describe these changes in your cover letter and we will update your Data Availability statement to reflect the information you provide.

4. PLOS requires an ORCID iD for the corresponding author in Editorial Manager on papers submitted after December 6th, 2016. Please ensure that you have an ORCID iD and that it is validated in Editorial Manager. To do this, go to ‘Update my Information’ (in the upper left-hand corner of the main menu), and click on the Fetch/Validate link next to the ORCID field. This will take you to the ORCID site and allow you to create a new iD or authenticate a pre-existing iD in Editorial Manager. Please see the following video for instructions on linking an ORCID iD to your Editorial Manager account: https://www.youtube.com/watch?v=_xcclfuvtxQ.

Reviewers' comments:

Reviewer's Responses to Questions

**Comments to the Author**

1. Is the manuscript technically sound, and do the data support the conclusions?

Reviewer #1: No

Reviewer #2: Partly

2. Has the statistical analysis been performed appropriately and rigorously? 

Reviewer #1: No

Reviewer #2: Yes

3. Have the authors made all data underlying the findings in their manuscript fully available?

Reviewer #1: No

Reviewer #2: Yes

4. Is the manuscript presented in an intelligible fashion and written in standard English?

Reviewer #1: Yes

Reviewer #2: No

5. Review Comments to the Author

Reviewer #1: This papier is important because this population of Ethiopa has neither been studied before.

The authors can in addition include the multicentric design of this study in 5 antenatal clinic.

The paper has a major the problem the absence of definition of psychological distress. Does the authors toke the severe, moderate symptoms according to the scale. They did not give the details according to sevreity symptoms.

Title should be change because the methodology of the study does’nt allow to evaluate predictive value of risk factors identified. To be predictive the factor should be evaluate at the begining of pregnancy or better before it and maternal distress evaluate before delivery of in post partum period.

Abstrat

Results add a s

Precise what is 174 ? number of patients ? please modify everywhere

The amount of non responder is very low. A such high rate is questionnable please comment.

Please give the details about maternal symptoms and thier intensity according with scale?

If IMC is avalaible it should be of interets

Discussion

There is a selection bias because women visiting clinic may have a different profils from other patients especially patients without antenatal visits. Please comments

How the five hospitals were selected ? why patients were seleted every 5 one ?

Generally the age and SD are give rather than age group of patients.

No need to give again the whole results with IC

In the discussion please discuss the scale use to evaluate the prevalence of maternal distress among countries. Is there a variation according the the scale use ? has it been already reported

Was the scale used in this study reproductible or already used in African countries. The authors in the methodology put this sentence « K10 scale has already been validated in Ethiopia by Tesfaye et al. and yielded an excellent internal consistency of 0.93, sensitivity of 84.2%, and specificity of 77.8% at a cut-off point of 6/7(33). Thus, it was reasonable to use for this study. In the methods just cite the reference remove the last sentence and move it into the discussion

population.

Authors wrote that the questionnaire was translate in different langages, does they find a difference among the local group of women and different langages?

The age of onset of most psychiatric disorders are early and also, psychological distress (anxiety and depression) are more common in female than men. Is this affirmation true ? please put a reference. There is no link between the two sentences and the results from this study give no information about the early age of onset of psychiatrics disorders nor the higher prevalence among women.

Please comment the absence of link between substance abuse, alcool and the sign of psychologic distress ?

Even though the link between abortion and mental illness is controversial, most women felt more regret, sadness and anger about the abortion than the pregnancy.I don’t understand the second sentence. Are this affirmation results from the study or information from previous study ?

Is there an Impact of religion on distress symptoms ?

Figure 2 Give no information. The substance abuse should be in the table and include in the multivariate analysis

Does the marital status and the unplanned delivery impacted the maternal health ?

Main concern is that the authors gave no information about the details of the scale results : Kessler Psychological Distress Scale-Kessler regarding intentisty of symptoms

A score 10 - 19 Likely to be well, 20 - 24 Likely to have a mild disorder, 25 - 29 Likely to have a moderate disorder, 30 - 50 Likely to have severed distress K10 scale is a 10-item questionnaire that a person rating the 30 days anxiety and depressive

symptoms experience in a five level Likert . prevalence of the results accordng with the scale should be ine the table and included in the statisticall analysis.

Reviewer #2: This study has scientific interest, but several important aspects should be reviewed by the

authors. I hope that my opinions will help shape your research article more precise and

interesting. The followings are my comments:

• On an overall level, and especially in the "Measures" section, I think that the language in which it is written should be revised, both in terms of grammar and punctuation.

• Both in the introduction and in the discussion, I consider that it would be optimal to insert the social determinants of health approach.

• In the section "Data collection procedures" it would be useful to indicate whether the questionnaires were self- or hetero-administered. Also, although this is explained in the final section on ethical statements, it would be useful to explain in this section how their consent to participate was collected, and to add how the study was explained to them and by whom, as well as how they were first contacted.

• In the section “Data quality control” it would be useful to explain the back translation procedure in more detail and to add citations and bibliographical references on this methodology.

• In the section “Data processing and analysis” explain whether and how the data were anonymized.

• Comments on the “Results”

o It would be clearer if the results were reported in the same order in which the data collection elements are described in the previous section.

o In the section “Obstetrics of the respondent” please provide the results in the same way throughout the section. Example of what appears in the manuscript: 352(46.4 %) of the study subjects / by (96)87.4 % of respondents.

• Comments on the “Discussion”

o There is a lack of bibliographical references to support what is described. Examples of contributions without references that appear in the manuscript:

That may be because more choices are made by educated people that they have greater control over their lives and better protection.

Even though the link between abortion and mental illness is controversial, most women felt more regret, sadness and anger about the abortion than the pregnancy.

In several studies the physical IPV pregnant women were more likely to experience mood disorders.

Social support provides physical and psychological advantages for people facing stressful physical and psychosocial events, and is considered a factor that reduces psychological distress when facing stressful events. In recent decades, numerous studies have been conducted on the impact of social support on health, quality of life and, in particular, mental health.

• In the section “Limitations” it would be appropriate to include the fact that some of the data collection instruments were not previously validated in the target population.

6. PLOS authors have the option to publish the peer review history of their article (what does this mean?). If published, this will include your full peer review and any attached files.

Reviewer #1: No

Reviewer #2: No

---

## [Author Response · Author response to Decision Letter 0]

14 Dec 2022

Manuscript ID number: 

PONE-D-22-01497

Title of paper:

Maternal psychological distress and associated factors among pregnant women attending antenatal care at public hospitals, Ethiopia

Response to reviewers’ and editor comments 

Dear editor and reviewers 

Author of this manuscript would like to thank all, for their constructive feedbacks and comments. I’m thankful for your timely response as well. Please kindly find submitted revised manuscript and point to point response to reviewers’ comments. 

With best regards,

Derebe Madoro

Corresponding author 

Responses for editor Comments

PONE-D-22-01497

Maternal psychological distress and its predictors among pregnant women attending antenatal care at public hospitals, Ethiopia

PLOS ONE

Dear Dr. Madoro,

Thank you for submitting your manuscript to PLOS ONE. After careful consideration, we feel that it has merit but does not fully meet PLOS ONE’s publication criteria as it currently stands. Therefore, we invite you to submit a revised version of the manuscript that addresses the points raised during the review process.

Author responses: Thank you so much dear editor, I checked for PLOS ONE's style requirements and edited accordingly.

Both reviewers agree on the need to submit a revised version of your manuscript. From my point of view, it is relevant that you include a definition of psychological distress, as well as address the different aspects that the reviewers point out, especially in the discussion and limitations sections.

Author responses: Thank you so much dear editor, definition of psychological distress included. Reviewers’ comments in the discussion and limitation section are addressed and indicated below in the reviewer response section.

Author responses: Thanks dear editor, the author(s) received no specific funding for this work. This included in the cover letter and figures are edited according to guidelines.

Author responses: laboratory protocols not applicable 

We look forward to receiving your revised manuscript.

Kind regards,

Patricia Moreno-Peral

Academic Editor

PLOS ONE

Author responses: Thank you so much dear editor, I checked for PLOS ONE's style requirements and edited accordingly

Author responses: Thank you so much dear editor, regarding data availability statement, Data cannot be shared because it may be replicated by others and because it may contain identifiable information. The data restriction was imposed by the Institutional Review Board. For further information, you can contact one of the committee members via this email address: seidshumye22@gmail.com. You can access the data upon request. On reasonable request, mobile: +251926023903, Dilla University, Dilla.. This also indicated in the revised cover letter.

4. PLOS requires an ORCID iD for the corresponding author in Editorial Manager on papers submitted after December 6th, 2016. Please ensure that you have an ORCID iD and that it is validated in Editorial Manager. To do this, go to ‘Update my Information’ (in the upper left-hand corner of the main menu), and click on the Fetch/Validate link next to the ORCID field. This will take you to the ORCID site and allow you to create a new iD or authenticate a pre-existing iD in Editorial Manager. Please see the following video for instructions on linking an ORCID iD to your Editorial Manager account: https://www.youtube.com/watch?v=_xcclfuvtxQ.

Author responses: ORCID iD is updated in the Editorial Manager as recommended 

Author responses: Thanks dear editor, ethics statement moved to method section.

Reviewers' comments:

Reviewer's Responses to Questions

Comments to the Author

1. Is the manuscript technically sound, and do the data support the conclusions?

Reviewer #1: No

Reviewer #2: Partly

2. Has the statistical analysis been performed appropriately and rigorously?

Reviewer #1: No

Reviewer #2: Yes

3. Have the authors made all data underlying the findings in their manuscript fully available?

Reviewer #1: No

Reviewer #2: Yes

4. Is the manuscript presented in an intelligible fashion and written in standard English?

Reviewer #1: Yes

Reviewer #2: No

5. Review Comments to the Author

 Responses for reviewer#1 comments

Reviewer #1: This papier is important because this population of Ethiopa has neither been studied before.

#The authors can in addition include the multicentric design of this study in 5 antenatal clinic.

The paper has a major the problem the absence of definition of psychological distress. Does the authors toke the severe, moderate symptoms according to the scale. They did not give the details according to sevreity symptoms.

 Author responses: Thanks dear reviewer for your constructive comments. Psychological distress is a general term for stress, anxiety, and depression symptoms. High levels of psychological distress can be a sign of common mental illnesses like depression and anxiety disorders and are a sign of poor mental health(1). Regarding details according to severity symptom, according to assumptions of logistic regression analysis the outcome variable should be dichotomous. Psychological distress was measured by Kessler Psychological Distress Scale (K10), which scored a total score of < 20 was considered normal; 20-24 mild distress; 25-29 moderate distress; and 30-50 severe distress. Therefore,in this study psychological distress categorized as below 20 normal and above 20 as positive for psychological distress. This edited in original manuscript and literature cited accordingly. 

#Title should be change because the methodology of the study does’nt allow to evaluate predictive value of risk factors identified. To be predictive the factor should be evaluate at the begining of pregnancy or better before it and maternal distress evaluate before delivery of in post partum period.

Author responses: Thank you so much dear reviewer, the title modified based on comments given by reviewer. It is replaced by ‘‘Maternal psychological distress and associated factors among pregnant women attending antenatal care at public hospitals, Ethiopia’’

Abstrat

#Results add as Precise what is 174 ? number of patients ? please modify everywhere The amount of non responder is very low. A high rate is questionnable please comment.

Author responses: Thank you dear reviewer for your comments. 

Regarding ‘174’, it is a number of patients positive for psychological distress, but we modified based on comments and to make it easy to understand we have put prevalence of maternal distress with percentage like 174(21.5%). Regarding low score of non-responders, A response rate was found to be 92%. Since the data was collected at each hospital of Antenatal care follow-up unit, most pregnant women not missed their hospital appointment and they were cooperative to participate in the study. Because of that reason getting non responders might be low. 

#Please give the details about maternal symptoms and thier intensity according with scale?

If IMC is avalaible it should be of interest

Author responses: Thank you dear reviewer for your comments. With regard to maternal symptoms and their intensity according with scale, as I mentioned above no ‘1’. Maternal psychological symptoms are described based on assumption of logistic regression, Normal and positive for maternal psychological symptoms. The scale put <20 as normal and 20 and above considered as having Maternal psychological distress. Dear reviewer, My apologies for I couldn’t understand the question and abbreviation of ‘IMC’ ‘‘If IMC is available it should be of interets’’ I can address it in my second revision.

Discussion

#There is a selection bias because women visiting clinic may have a different profils from other patients especially patients without antenatal visits. Please comments

Author responses: Thank you so much, regarding selection bias, my study population is not all women visiting clinic, but my study population was only pregnant women who visit selected public hospitals, therefore no selection bias seen, because other patients are not part of our study. Regarding the hospitals selection, the 5 public hospitals were selected based on giving antenatal care services located in Addis Ababa. 

#How the five hospitals were selected ? why patients were seleted every 5 one ?

Generally the age and SD are give rather than age group of patients. 

Author responses: Participants were selected every five, because we have used systematic sampling technique to select participants. To fill the assumption of systematic sampling, we must calculate ‘k’ and use every kth interval to select study participants. The age has already putted with standard deviation.

#No need to give again the whole results with IC

Author responses: I think ‘IC’ miss spelt, if it is CI(confidence interval), The reason to put a result with CI, to make it clear for readers to identify easily upper and lower limit to compare the result of this study with other study results whether reported above or below the mentioned prevalence in this study. 

#In the discussion please discuss the scale use to evaluate the prevalence of maternal distress among countries. Is there a variation according the the scale use ? has it been already reported

Author responses: Thank you so much for this comment, Regarding variation of scale, this study reported study participants as positive and negative for psychological distress, we discussed it accordingly. That means not reported based on the severity scale. Since the analysis was logistic regression, the outcome variable should be dichotomized (yes/no). 

#Was the scale used in this study reproductible or already used in African countries. The authors in the methodology put this sentence « K10 scale has already been validated in Ethiopia by Tesfaye et al. and yielded an excellent internal consistency of 0.93, sensitivity of 84.2%, and specificity of 77.8% at a cut-off point of 6/7(33). Thus, it was reasonable to use for this study. In the methods just cite the reference remove the last sentence and move it into the discussion.

Author responses: Thanks dear reviewer, Yes, the psychological distress scale (k10) was validated and already used in Ethiopia/Africa and the reference cited already as recommended by reviewer. 

#Authors wrote that the questionnaire was translate in different langages, does they find a difference among the local group of women and different langages?

Author responses: Thank you so much, There was no difference seen among different languages. The translation was done to control the quality of data and to check for consistency and understandability of the tool before the actual data was collected.

#The age of onset of most psychiatric disorders are early and also, psychological distress (anxiety and depression) are more common in female than men. Is this affirmation true ? please put a reference. There is no link between the two sentences and the results from this study give no information about the early age of onset of psychiatrics disorders nor the higher prevalence among women.

Author responses: Thanks once again for interesting comments, a sentences ‘‘psychological distress (anxiety and depression) are more common in female than men’’ is removed because not in line with the findings in the discussion. Regarding age of onset, study revealed that with increasing age, psychological distress generally declined across the age range 20-64 years. The age of onset of most psychiatric disorders are early and also, psychological distress (anxiety and depression) in pregnancy(45). 

#Please comment the absence of link between substance abuse, alcool and the sign of psychologic distress ?

Author responses: Thank you so much dear reviewer. The reason for absence of link between substance use, alcohol and psychological distress could be those using substance like alcohol, used small amount due to fear of negative effect of substances on pregnancy and may not develop substance use disorder. In this case being a risk to have psychological distress could be lower. In this study we have assessed substance use like alcohol, cigarettes, chat and other substance, but didn’t assess substance use disorder. Another reason might be their coping strategy differences, because some of individuals use substance as coping mechanism. 

#Even though the link between abortion and mental illness is controversial, most women felt more regret, sadness and anger about the abortion than the pregnancy.I don’t understand the second sentence. Are this affirmation results from the study or information from previous study ? Is there an Impact of religion on distress symptoms ?

Author responses: Thank you so much dear reviewer. Regarding abortion and mental health, the second sentences taken from the finding from comprehensive review study not the result of this study and the reference is cited accordingly. 

Regarding religion, even though religion is not significantly associated in this study, it could have impact on psychological distress symptoms. But religion is not significantly associated in this study.

#Figure 2 Give no information. The substance abuse should be in the table and include in the multivariate analysis

Author responses: Thank you so much. I added percentage for each substance in the figure to make it more informative. Out of all substances included in this study those score a p-value<0.25 in the bi-variable logistic regression analysis were entered into the multivariate logistic regression model. Current alcohol and cigarette use were two substances included in to multivariate analysis. But others are did not fulfill the above criteria.

#Does the marital status and the unplanned delivery impacted the maternal health ?

Author responses: Thank you so much. With regard to marital status and unplanned delivery, even though, they didn’t have significant association in this study, they could have impact on the maternal health. 

#Main concern is that the authors gave no information about the details of the scale results : Kessler Psychological Distress Scale-Kessler regarding intentisty of symptoms

A score 10 - 19 Likely to be well, 20 - 24 Likely to have a mild disorder, 25 - 29 Likely to have a moderate disorder, 30 - 50 Likely to have severed distress K10 scale is a 10-item questionnaire that a person rating the 30 days anxiety and depressive

symptoms experience in a five level Likert . prevalence of the results accordng with the scale should be ine the table and included in the statisticall analysis.

Author responses: Thanks again for constructive comments, Regarding psychological distress scale as I mentioned above no ‘1’, since I used logistic regression analysis, the assumption don’t allow me to use severity scale in analysis. The prevalence and statistical analysis could not be presented with psychological distress scale, unless applying linear regression analysis. Therefore, in this study I dichotomized outcome variable as presence (those scored 20 and above) and absence (those scored below 20) of psychological distress as applied in different study. The missed point is included in the original manuscript.

Responses for reviewer#2 comments

Reviewer #2: This study has scientific interest, but several important aspects should be reviewed by the authors. I hope that my opinions will help shape your research article more precise and interesting. The followings are my comments:

• On an overall level, and especially in the "Measures" section, I think that the language in which it is written should be revised, both in terms of grammar and punctuation.

Author responses: Thank you dear reviewer for your comments. We have tried to address and edit the language aspect of this manuscript by professionals from linguistic department

• Both in the introduction and in the discussion, I consider that it would be optimal to insert the social determinants of health approach.

Author responses: Dear reviewer thank you so much, regarding social determinants of health approach, I added and discussion was done for significantly associated factors accordingly. The social determinants influencing maternal health were poverty, low socio-economic status, decreasing age, low maternal educational status, abortion history, violence during pregnancy and single status are some of the factors for pregnancy with psychological distress(11-14). The main mediating variables were location, maternal age at childbirth, parity, women's media exposure, and maternal health messages(15).

• In the section "Data collection procedures" it would be useful to indicate whether the questionnaires were self- or hetero-administered. Also, although this is explained in the final section on ethical statements, it would be useful to explain in this section how their consent to participate was collected, and to add how the study was explained to them and by whom, as well as how they were first contacted.

Author responses: Dear reviewer thank you so much, Regarding the questionnaire, data was collected using face to face interview with interviewer administered questionnaire.

Participants were first contacted with data collectors after permission letter has received from each selected hospital managers. Written informed consent was secured from each participant during study period. Participants were informed by the data collectors about the aim of the study and no names were recorded to maintain confidentiality. The participants were informed of their right to refuse or stop participating at any time during the interview.

• In the section “Data quality control” it would be useful to explain the back translation procedure in more detail and to add citations and bibliographical references on this methodology.

Author responses: Thank you dear reviewer for your comments. The entire questionnaire was translated into local languages Amharic then it was translated back to English by an independent person to check for consistency and understandability of the tool. A back translation allows comparing translations with the original text for quality and accuracy. It also helps to evaluate equivalence of meaning between the source and target texts(37) 

• In the section “Data processing and analysis” explain whether and how the data were anonymized.

Author responses: Dear reviewer thank you so much, removed direct identifiers and replaced potentially disclosive free-text responses with more general text to assure data was anonymized. 

• Comments on the “Results”

o It would be clearer if the results were reported in the same order in which the data collection elements are described in the previous section.

Author responses: Thank you so much, As comments given, the results ordered according to data collection elements. Thanks in advance 

o In the section “Obstetrics of the respondent” please provide the results in the same way throughout the section. Example of what appears in the manuscript: 352(46.4 %) of the study subjects / by (96)87.4 % of respondents.

Author responses: Edited as comments given, thanks dear reviewer‘‘352(46.4 %) of the study subjects were multiparous and 642(84.6%) of the pregnancies were planned. Besides, history of abortion was experienced by 96(87.4 %) of respondents.’’

• Comments on the “Discussion”

o There is a lack of bibliographical references to support what is described. Examples of contributions without references that appear in the manuscript:

That may be because more choices are made by educated people that they have greater control over their lives and better protection.

Author responses: Reference is cited, thanks for comments

Even though the link between abortion and mental illness is controversial, most women felt more regret, sadness and anger about the abortion than the pregnancy.

Author responses: Reference is cited, thanks for comments

In several studies the physical IPV pregnant women were more likely to experience mood disorders.

Author responses: Reference is cited, thanks for comments

Social support provides physical and psychological advantages for people facing stressful physical and psychosocial events, and is considered a factor that reduces psychological distress when facing stressful events. In recent decades, numerous studies have been conducted on the impact of social support on health, quality of life and, in particular, mental health.

Author responses: Reference is cited, thanks for comments

• In the section “Limitations” it would be appropriate to include the fact that some of the data collection instruments were not previously validated in the target population.

Author responses: Thanks in advance dear reviewer, the comment added as recommended

6. PLOS authors have the option to publish the peer review history of their article (what does this mean?). If published, this will include your full peer review and any attached files.

Do you want your identity to be public for this peer review? For information about this choice, including consent withdrawal, please see our Privacy Policy.

Reviewer #1: No

Reviewer #2: No

Attachments area

---

## [Editor Report · Decision Letter 1]

2 Jan 2023

Maternal psychological distress and associated factors among pregnant women attending antenatal care at public hospitals, Ethiopia

PONE-D-22-01497R1

Dear Dr. Derebe Madoro,

We’re pleased to inform you that your manuscript has been judged scientifically suitable for publication and will be formally accepted for publication once it meets all outstanding technical requirements.

Kind regards,

Patricia Moreno-Peral

Academic Editor

PLOS ONE

---

## [Editor Report · Acceptance letter]

8 Jan 2023

PONE-D-22-01497R1 

Maternal psychological distress and associated factors among pregnant women attending antenatal care at public hospitals, Ethiopia 

Dear Dr. Madoro:

I'm pleased to inform you that your manuscript has been deemed suitable for publication in PLOS ONE. Congratulations! Your manuscript is now with our production department. 

Kind regards, 

on behalf of

Dr. Patricia Moreno-Peral 

Academic Editor

PLOS ONE